# Clinical courses, impact and prognostic indicators for a persistent course of low back pain: Results from a population-based cohort study

**Luís Antunes Gomes**[1,2,3]\*, **Ana Maria Rodrigues**[1,2,4], **Jaime C. Branco**[1,2,5], **Helena Canhão**[1,2,6], **Eduardo Brazete Cruz**[1,3]

**1** CHRC, Comprehensive Health Research Centre, NOVA Medical School, Universidade NOVA de Lisboa, Lisbon, Portugal, **2** EpiDoC Unit, NOVA Medical School, Universidade NOVA de Lisboa, Lisbon, Portugal, **3** Departamento de Fisioterapia, Escola Superior de Saúde, Instituto Politécnico de Setúbal, Setubal, Portugal, **4** Unidade de Reumatologia, Hospital dos Lusíadas, Lisbon, Portugal, **5** Serviço de Reumatologia do Hospital Egas Moniz—Centro Hospitalar Lisboa Ocidental (CHLO-EPE), Lisbon, Portugal, **6** Unidade de Reumatologia, Centro Hospitalar Universitário Lisboa Central–Hospital Santo António dos Capuchos (CHULC-EPE), Lisbon, Portugal

\* ftluisgomes@gmail.com

## Abstract

### Background

Low back pain (LBP) is a long-term health condition with distinct clinical courses. Its characterization together with the identification of prognostic factors for a persistent LBP course may trigger the development of personalized interventions. This study aimed to investigate the courses of chronic LBP (CLBP), its cumulative impact, and the indicators for the persistence of pain.

### Material and methods

Patients with active CLBP from the EpiDoC, a population-based cohort study of a randomly recruited sample of 10.661 adults with prolonged follow-up, were considered. Pain, disability, and health-related quality of life (HRQoL) were assessed at three time-points over five years. According to their pain symptoms over time, participants were classified as having a persistent (pain at the baseline and at all the subsequent time-points) or a relapsing pain course (pain at the baseline and no pain at least in one of the subsequent time-points). A mixed ANOVA was used to compare mean differences within and between patients of distinct courses. Prognostic indicators for the persistent LBP course were modulated through logistic regression.

### Results

Among the 1.201 adults with active CLBP at baseline, 634 (52.8%) completed the three time-points of data collection: 400 (63.1%) had a persistent and 234 (36.9%) a relapsing course. Statistically significant interactions were found between the group and time on

**Data Availability Statement:** All relevant data are within the manuscript and its Supporting Information files.

**Funding:** The present publication was supported by national funds through FCT – Fundação para a Ciência e a Tecnologia, I. P., under the MyBack project (PTDC/SAU-SER/7406/2020), in which Eduardo Brazete Cruz is the principal investigator. The corresponding author, Luís Antunes Gomes, is supported by an individual PhD grant (SFRH/BD/145636/2019) of the same institution.

**Competing interests:** The authors have declared that no competing interests exist.

disability ($F(2,1258) = 23.779$, $p<0.001$) and HRQoL ($F(2,1252) = 82.779$, $p<0.001$). In the adjusted model, the persistent course was associated with the disability level (OR 1.86, CI95% 1.40–2.40, $p<0.001$), depressive symptoms (OR 1.96, CI95% 1.21–3.18, $p = 0.007$), female gender (OR 1.90, CI95% 1.26–2.87, $p = 0.002$) and having a manual job (OR 1.46, CI95% 1.02–2.10, $p = 0.040$).

## Conclusion

In the long-term, patients with CLBP may follow a persistent or relapsing course of pain. Being female, presenting depressive symptoms, having a manual job and higher disability at baseline predicts a persistent course of LBP.

## Introduction

Non-specific low back pain (LBP) is one of the most common health problems in societies and is strongly associated with poor self-rated general health [1–4]. It causes long-term pain, disability, work absenteeism, and increased use of healthcare services [3, 4].

LBP is traditionally divided into acute (less than 6 weeks), sub-acute (6 to 12 weeks), or chronic (more than 12 weeks) based on symptom duration [5]. The typical course of acute is initially favourable, i.e., is characterized by a marked reduction in the mean of pain and disability in the first 6 weeks [6]. However, from this point forward, the improvement slows down and the probability of developing a persistent back pain condition (CLBP–Chronic Low Back Pain) increases for approximately 40% of the patients [6, 7]. Some of these patients will recover, but a minority will develop a persistent, disabling condition. This minority is responsible for most of the social and health costs associated with LBP [8], to which absenteeism and reduced productivity costs must be added [9].

To better address the individual consequences of persistent, disabling LBP on patients and eventually define more appropriate treatment solutions, an improved understanding of its long-term prognosis is needed. Traditionally, LBP episodes have been understood and examined as episodic, recurrent but unrelated [10, 11]. Thus, most of the published research is focused on the evaluation of a single episode or presence of pain or disability at specific time-points, making it difficult to know whether a particular study includes individuals with an initial episode of LBP or/and individuals with an exacerbation of a recurrent LBP condition [12, 13].

The population-based studies with long-term follow-up (5 to 10 years) published to date have been contesting the traditional classification of LBP, suggesting that more than an episodic condition of unrelated episodes, LBP is a long-term condition with a variable course of pain and multiple interrelated episodes [14–21]. Despite differences in the case definition or in the assessment time-points, only 6 to 14% of adults enrolled in these studies have a persistent course of LBP, with pain at all specific time-points of the follow-up [18, 19]. In contrast, 23 to 35% of the population studied reported that they were "painless at the time of evaluation" [16–18] or had "no day of pain during the previous year" [20]. These results suggest that there is a considerable percentage of individuals reporting pain at a given time who fully recover, while others maintain some level of pain and disability over time, with a small subgroup reporting a persistent and potentially disabling course of pain [16, 18–20]. Consequently, the clinical course based on the population means may not adequately reflect the within-person variation in LBP, opening the possibility of classifying patients into clinically relevant subgroups [16, 19].

The Epidemiology of Chronic Diseases Cohort Study (EpiDoC), being a population-based study with prolonged follow-up [22–24], produced a set of data that could help identify distinct LBP courses and investigate their impact on disability and health-related quality of life (HRQoL). On the other hand, the presence of distinct clinical courses among LBP patients opens the possibility of identifying prognostic indicators for a persistent LBP course. Although a set of indicators has been associated with the development of persistent, disabling pain such as disability, low levels of HRQoL, maladaptive pain coping behaviours, or depressive symptoms [25–30], the majority have emerged from studies that have evaluated pain at a single point of time, with follow-up limited to one year and based on heterogenous samples of patients regarding the previous history of LBP [31, 32]. The identification of predictors for the most disabling LBP course might allow healthcare professionals to better identify the patients at high-risk of persistent, disabling symptoms, and eventually, change their long-term prognosis through forms of treatment that best fit and respond to their characteristics [16].

The aims of this study were 1) to investigate the presence of different courses of pain across five years in a sample of patients with active CLBP; 2) to examine their cumulative impact on disability and HRQoL and; 3) to identify prognostic indicators for the persistent LBP course.

## Materials and methods

### Data source and study design

This study used longitudinal data of participants with active CLBP collected under the scope of the EpiDoC, a prospective cohort study that began with the EpiDoC 1 (EpiReumaPt) in 2011–13. EpiReumaPt was a national, cross-sectional, population-based study with a representative sample of the Portuguese population, comprising 10.661 non-institutionalized adults (≥18 years), living in Portugal and able to speak and read the Portuguese language [22, 23]. The recruitment of participants followed a three-stage approach. Aiming to characterize the Portuguese adult population and screen for the presence of rheumatic and musculoskeletal diseases (RMDs), the selection of participants for the first stage of the study (RMDs screening) followed a multistage random sampling, taking into account the stratification of the Portuguese statistic administrative territorial units (Nomenclature of Territorial Units for Statistics) in the 2001 CENSUS and the size of the population. Using a random route methodology, face-to-face interviews to randomly select individuals' households were conducted by a team of interviewers (i.e., non-physicians trained for this purpose). In the second stage (RMD diagnosis), both individuals who screened positive for at least one RMD and a random sample of 20% of individuals free of RMDs were evaluated by a rheumatologist so a diagnosis could be established. In the third and last stage (RMDs diagnostic validation), a team of experienced rheumatologists validated the previous RMD diagnosis. All participants recruited for the first stage of EpiDoC 1 were invited to participate in the following waves of the study; 10.153 (95.2%) agreed and signed the consent forms. From there, and over a 5-year period, two subsequent timepoints of data collection (every 18 months) have been completed, through telephone interviews, using data based on the same participants [24]. To prevent withdrawals, when a participant was not available, a maximum of 6 additional attempts were made at different times [24]. The detailed procedures for the study design and sample selection and recruitment are published elsewhere [23, 24]. The EpiDoC followed the principles established by the Declaration of Helsinki and revised in 2013 in Fortaleza. The study was approved by the Ethics Committee of NOVA Medical School and the Portuguese Data Protection Authority. All participants gave written informed consent [22, 23].

## Study sample

The study sample consisted of all the individuals recruited under the EpiReumaPt who self-reported active CLBP on the day of the interview [4]. Active CLBP was defined as pain located in the lower back, between the lower thorax and the gluteal folds, that was present most of the time for a minimum period of 90 days, and without a specific cause. Individuals with a diagnosis of other spinal pain (neck or dorsal back pain) were excluded [4].

For this study, a more stringent case definition was used. Participants who reported "no pain/discomfort" at baseline when answering question 4 of the Euroqol, 5 dimensions, 3 levels (EQ-5D-3L) *(Regarding pain/discomfort please indicate which statement best describes your health today*?), were excluded. Finally, only those participants who participated in all the time-points of data collection (baseline and time-points 2 and 3) were included in the final analysis.

## Outcomes measures

At baseline, a standardized questionnaire was used to collect both basic sociodemographic and clinical characteristics and self-reported instruments of disability, HRQoL, and anxiety and depressive symptoms. At the follow-up time-points, the same self-reported measures of disability and HRQoL were used. Disability was measured using the Portuguese version of the Health Assessment Questionnaire (HAQ), a 20-question instrument that evaluates eight categories of functioning (dressing, rising, eating, walking, hygiene, reach, grip, and usual activities) over the previous week. Each category was scored from 0 (no disability) to 3 points (completely disabled). The final score also varied between 0 and 3 points, with higher scores corresponding to higher disability [33]. HRQoL was measured with the Portuguese version of EQ-5D-3L. This instrument comprises five questions regarding different dimensions of HRQoL (mobility, self-care, usual activities, pain/ discomfort, and anxiety/ depression) which are scored on a three-level severity scale (0 = "no problem"; 1 = "moderate problem", or 2 = "extreme problem"). The overall index of health status ranges between 0 ("death") and 1 ("the best possible health"). The weight applied to the severity states of the EQ-5D-3L was based on the Portuguese valuation study of the EQ-5D-3L [34]. Both HAQ and EQ-5D-3L instruments have been cross-culturally validated for the Portuguese language and showed good to excellent psychometric properties [35, 36].

## Prognostic indicators

The selection of potential prognostic indicators for the persistent LBP course was based on previously published literature [25–30] and covered four domains: socio-demographic (age, gender, employment status, marital status, and education), lifestyle (body mass index [BMI], physical exercise, and its frequency), psychosocial (anxiety and depression) and health-related indicators (pain, disability and HRQoL).

Anxiety and depression were measured through the Portuguese version of the Hospital Anxiety and Depression Scale (HADS) which is divided into an Anxiety subscale (HADS-A) and a Depression subscale (HADS-D) (the cut-off used for positive anxiety and depression symptoms was >11). The HADS has been cross-culturally validated in the Portuguese language and shown good to excellent psychometric properties [37, 38].

## Statistical analysis

All data analyses were performed using SPSS, Version 24.0 for OX Yosemite (SPSS Inc., Chicago, IL).

Firstly, since complete case analysis may lead to biased results because data are unlikely to be missing completely at random [39], nonresponse bias was assessed by comparing the characteristics of the respondent and non-respondent samples at baseline (time-point 1). For this analysis, responders were those who fulfilled the eligibility criteria, while non-responders were those who, despite the presence of both CLBP and baseline pain measured by question 4 of the EQ-5D-3L, had not participated in all time-points of data collection. A chi-square test and a students' *t*-test were used to compare the demographic and clinical data of responders and non-responders, as appropriate.

Secondly, different courses of LBP were represented by a three-word variable based on the participants' answer to question 4 of the EQ-5D-3L: from left to right, the first word stands for pain status at baseline, while the second and the third for the pain status at time-point 2 and 3, respectively. Since every participant had "moderate or extreme pain/discomfort" (pain dimension of the EQ-5D-3L) at baseline, courses of pain were defined based on the pain status of subsequent time-points. Two different courses were expected: a "persistent LBP course" for participants who reported "moderate or extreme pain/discomfort" in both time-points 2 and 3 (pain/pain/pain); and a "relapsing course" for those reporting "no pain/discomfort" at least in one time-point (pain/pain/no pain; pain/no pain/pain or; pain/no pain/no pain). Descriptive and inference statistics, namely the chi-square test and the independent *t*-test, were used to characterize and contrast LBP courses, respectively. Within and between groups differences in disability and HRQoL, over the three time-points, were examined using a mixed ANOVA (General Linear Model procedure). Aiming to identify which pairs of means differed from each other, for the interaction between groups and time-points, a post-hoc test comprising multiple comparison testing with Bonferroni correction was performed.

Finally, to identify prognostic indicators, associations between the baseline variables and the persistent LBP course were modelled through binary logistic regression analysis. To optimize the clinical interpretation of the results, some answer variables of the potential prognostic indicators were merged: education level (categorized as less vs. 10 or more years of education); marital status (living alone vs. living with someone); employment status (working vs. not working); BMI (underweight/ normal weight vs. overweight/ obesity); and pain (moderate vs. extreme pain). Multiple logistic regression was used through the backward conditional procedure. All comparisons were adjusted for variables that reveal statistically significant differences in bivariate analysis ($p<0.20$). The non-significant variable with the highest *p*-value was removed in a stepwise manner until all variables had $p<0.05$. The results are presented as crude and adjusted odds ratios (OR) with 95% confidence intervals. The discriminative ability of the model was assessed through the area under the receiver operating characteristic curve (AUC), ranging from 0.5 (no discrimination) to 1.0 (perfect discrimination) [40]. All *p* values are two-sided, and the significance level was 5%.

## Results

Of the 1.201 selected participants with active CLBP at baseline (time-point 1), 634 (52.8%) completed all the time-points of data collection (complete data on main outcomes–EQ-5D-3L index and HAQ). Non-responders at time-point 2 and/or 3 were lost due to unsuccessful contact and for other reasons (participants refused to sign the consent form for follow-up, expressed the wish to leave the study, had an invalid contact, or died) (Fig 1).

At baseline, no statistically significant differences in socio-demographic and clinical characteristics were found between responders and non-responders, except for HRQoL (EQ-5D-3L index, $p = 0.005$), with responders reporting a statistically significantly lower HRQoL. The

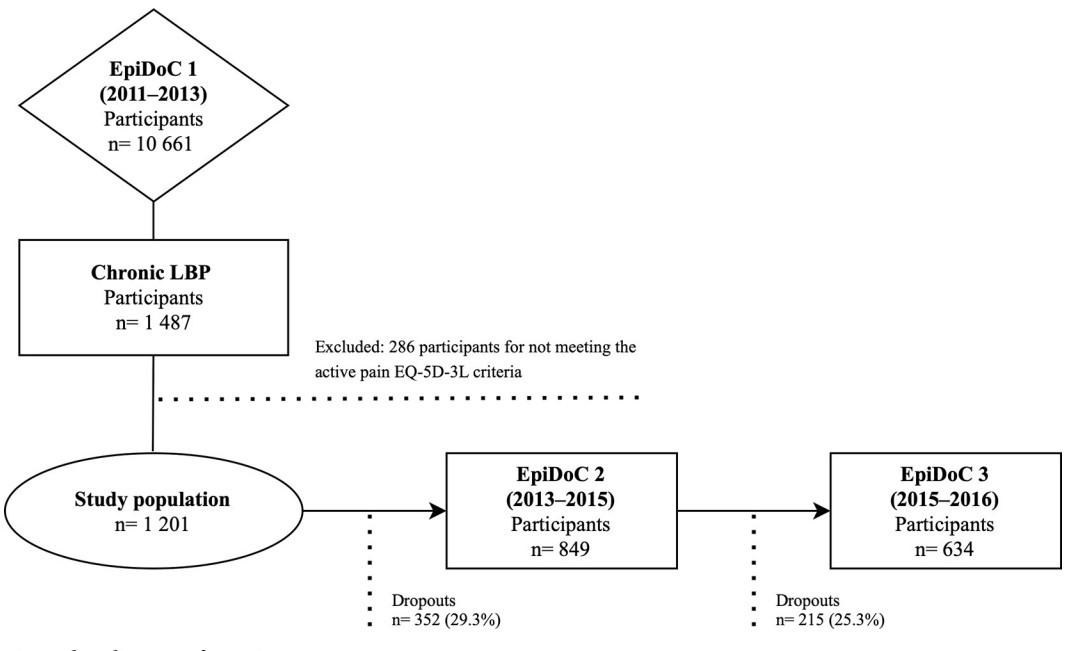

**Fig 1. Flow diagram of recruitment.**

participants were aged 61.6 ± 13.6 years, predominantly females (78.4%) and with a low educational level (64.4%).

## Courses of LBP

Considering the three time-points of data collection, 400 (63.1%) participants were classified as having a persistent LBP course and 234 (36.9%) as having a relapsing LBP course. Of the 234 participants classified with a relapsing course, 63 (26.9%) reported no pain in time-points 2 and 3 (pain/no pain/no pain), 49 (20.9%), an intermittent pattern of pain (pain/no pain/pain), and 122 (52.1%) reported no pain only in the third time-point (pain/pain/no pain). The characteristics of the relapsing and persistent groups are presented in Table 1.

Statistically significant differences were found between groups for age ($p = 0.008$), educational level ($p = 0.001$), HRQoL (EQ-5D-3L Index, $p = 0.005$), and disability ($p = 0.005$). The participants in the persistent group were older and less educated, had higher levels of disability, and had lower HRQoL. Moreover, statistically significant associations were found between gender ($p \leq 0.005$), anxiety ($p = 0.024$) and depressive symptoms ($p \leq 0.005$), and the course of LBP (relapsing or persistent).

## The course of disability and HRQoL between groups, over time

HRQoL and disability status have improved over time in the relapsing group, whereas in the persistent group they have remained relatively stable (Table 2). Statistically significant interactions were found between the group (relapsing or persistent) and time (time-point 1 to 3) on disability (HAQ, $F(2,1258) = 23.779$, $p<0.001$), and HRQoL (EQ-5D-3L index, $F(2,1252) = 82.779$, $p<0.001$). At any time-point there were statistically significant differences in disability (HAQ, $p<0.005$) and HRQoL (EQ-5D-3L index, $p<0.005$), between groups. Moreover, an opposite pattern of impact was found between groups; compared to baseline, HRQoL was significantly higher (EQ-5D-3L, $p<0.001$), and disability was significantly lower (HAQ, $p<0.001$) at time-point 3 in the relapsing group, whereas HRQoL was significantly lower (EQ-5D-3L,

**Table 1. Socio-demographic characteristics of the participants at baseline.** Total sample and groups (n = 634).

| | Total sample (n = 634) | Relapsing LBP course (n = 234) | | Persistent LBP course (n = 400) | |
|---|---|---|---|---|---|
| | n (%) | n | (%) | n | (%) |
| **Age** (years), mean ± SD | 61.6 ± 13.6 | 59.6 ± 15.1 | | 62.7 ± 12.7 | |
| **Gender** (female) | 497 (78.4%) | 158 | 31.8% | 339 | 68.6% |
| **Educational level** | | | | | |
| 0–4 years | 408 (64.4%) | 128 | 31.4% | 280 | 68.6% |
| 5–9 years | 113 (17.8%) | 56 | 49.6% | 57 | 50.4% |
| 10–12 years | 65 (10.3%) | 23 | 35.4% | 42 | 64.6% |
| >12 years | 37 (5.8%) | 22 | 59.5% | 15 | 40.5% |
| Do not know/do not answer | 11 (1.7%) | 5 | 45.5% | 6 | 54.5% |
| **Marital status** | | | | | |
| Single | 41 (6.5%) | 17 | 41.5% | 24 | 58.5% |
| Married | 400 (63.1%) | 144 | 36.0% | 256 | 64.0% |
| Divorced | 38 (6.0%) | 15 | 39.5% | 23 | 60.5% |
| Widower | 144 (22.7%) | 54 | 37.5% | 90 | 62.5% |
| Consensual union | 11 (1.7%) | 4 | 36.4% | 7 | 63.6% |
| **Anxiety positive symptoms** | 194 (30.6%) | 59 | 30.4% | 135 | 69.6% |
| **Depressive positive symptoms** | 135 (21.3%) | 28 | 20.7% | 107 | 79.3% |
| **Employment status** | | | | | |
| Full-time employed | 123 (19.4%) | 57 | 46.3% | 66 | 53.7% |
| Part-time employed | 13 (2.1%) | 6 | 46.2% | 7 | 53.8% |
| Domestic worker | 61 (9.6%) | 18 | 29.5% | 43 | 70.5% |
| Unemployed | 49 (7.7%) | 22 | 44.9% | 27 | 55.1% |
| Retired | 344 (54.3%) | 117 | 34.0% | 227 | 66.0% |
| Student | 2 (0.3%) | 1 | 50.0% | 1 | 50.0% |
| Temporary disabled | 21 (3.3%) | 7 | 33.3% | 14 | 66.7% |
| Do not know/do not answer | 21 (3.3%) | 6 | 28.6% | 15 | 71.4% |
| **Manual worker (Yes)** | 397 (62.6%) | 133 | 33.5% | 264 | 66.5% |
| **BMI (kg/m²)** | n = 588 | n = 221 | | n = 367 | |
| Underweight (<18.50 kg/m²) | 1 (0.2%) | 1 | 100.0% | 0 | 0.0% |
| Normal (18.5–24.99 kg/m²) | 162 (27.6%) | 70 | 43.2% | 92 | 56.8% |
| Overweight (15.00–29.99 kg/m²) | 234 (39.8%) | 90 | 38.5% | 144 | 61.5% |
| Obese (≥30.00 kg/m²) | 191 (32.5%) | 60 | 31.4% | 131 | 68.6% |
| **Physical exercise** (Yes) | 134 (21.1%) | 51 | 38.1% | 83 | 61.9% |
| **Average time per day** (min) | n = 134 | n = 51 | | n = 83 | |
| ≤45 min | 63 (47.0%) | 22 | 34.9% | 41 | 65.1% |
| >45 min | 71 (53.0%) | 29 | 40.8% | 42 | 59.2% |
| **Average time per week** (min) | n = 134 | n = 51 | | n = 83 | |
| ≤150 min/week | 61 (45.5%) | 22 | 36.1% | 39 | 63.9% |
| >150 min/week | 73 (54.5%) | 29 | 39.7% | 44 | 60.3% |
| **Pain (question 4—EQ-5D-3L)** | | | | | |
| Moderate pain or discomfort | 547 (86.3%) | 209 | 38.2% | 338 | 61.8% |
| Extreme pain or discomfort | 87 (13.7%) | 25 | 28.7% | 62 | 71.3% |
| **Disability (HAQ, 0–3),**[a] mean ± SD (n = 628) | 0.99 ± 0.72 | 0.76 ± 0.69 | | 1.13 ± 0.70 | |
| **HRQoL (EQ-5D-3L index, 0–1),**[b] mean ± SD | 0.46 ± 0.19 | 0.52 ± 0.19 | | 0.43 ± 0.19 | |

BMI, body mass index; EQ-5D-3L, euroqol, 5 dimensions, 3 levels; HAQ, health assessment questionnaire; HRQoL, health-related quality of life; LBP, low back pain; min, minutes; SD, standard deviation.

[a] HAQ, 0–3: A lower HAQ disability score indicates better function.

[b] EQ-5D-3L index, 0–1: "1" corresponds to the best possible health, and "0" to death. Negative values correspond to states of health considered to be worse than death.

**Table 2. HRQoL and disability at the different time-points, in "relapsing" and "persistent" LBP groups (n = 634).**

| | | Mean ± SD Relapsing LBP course | Mean ± SD Persistent LBP course | *p*-value |
|---|---|---|---|---|
| Time-point 1 (Baseline) | EQ-5D-3L index, 0–1[a] (n = 628) | 0.52 ± 0.19 | 0.43 ± 0.19 | <0.001 |
| | HAQ, 0–3[b] (n = 634) | 0.76 ± 0.69 | 1.13 ± 0.70 | <0.001 |
| Time-point 2 | EQ-5D-3L index, 0–1[a] (n = 634) | 0.58 ± 0.28 | 0.30 ± 0.23 | <0.001 |
| | HAQ, 0–3[b] (n = 631) | 0.38 ± 0.55 | 0.69 ± 0.80 | <0.001 |
| Time-point 3 | EQ-5D-3L index, 0–1[a] (n = 634) | 0.70 ± 0.32 | 0.34 ± 0.24 | <0.001 |
| | HAQ, 0–3[b] (n = 631) | 0.55 ± 0.60 | 1.27 ± 0.70 | <0.001 |

EQ-5D-3L, euroqol, 5 dimensions, 3 levels; HAQ, health assessment questionnaire; HRQoL, health-related quality of life; LBP, low back pain; SD, standard deviation.

[a] HAQ, 0–3: A lower HAQ disability score indicates better function.

[b] EQ-5D-3L index, 0–1: "1" corresponds to the best possible health, and "0" to death. Negative values correspond to states of health considered to be worse than death.

*p*<0.001), and disability was significantly higher (HAQ, *p*<0.001) at time-point 3 in the persistent group.

### Prognostic indicators for the persistent LBP course

After adjustments, a persistent course of pain was associated with disability levels, depressive symptoms, being female and having a manual job at the baseline. Participants with higher levels of disability at the baseline had a higher odds ratio (OR) for the persistence of pain (OR 1.86, CI95% 1.40–2.40, *p*<0.001). Participants with positive depressive symptoms, manual workers, and females have also a higher probability of developing a persistent LBP course than non-depressive participants (OR 1.96, CI95% 1.21–3.18, *p* = 0.007), non-manual workers (OR 1.46, CI95% 1.02–2.10, *p* = 0.040), and males (OR 1.90, CI95% 1.26–2.87, *p* = 0.002) (Table 3).

### Discussion

This study aimed to identify different courses of pain within a sample of patients with active CLBP during a follow-up period of five years; to examine their cumulative impact on disability and HRQoL and; to identify potential predictors for the persistence of symptoms. To our knowledge, this was the first large prospective study that addresses the long-term course and prognosis of a restricted sample of patients with active CLBP. Notwithstanding CLBP has been traditionally and simplistically described as a stable condition of persistent symptoms where remission is unlikely, the present results support the current understanding of LBP as a relapsing condition of interrelated episodes of pain [11, 14, 18–20, 41].

In this study, 36.9% of participants reported at least one time-period of complete remission of LBP symptoms. This result is consistent with other long-term population-based studies' which suggest that 23 to 35% of individuals are free of pain in all follow-up time-points. In contrast, the proportion of participants reporting pain at all time-points (63.1%) was substantially higher than in previous studies whose values ranged between 6 and 29% [18–20]. These differences might be in part explained by differences in sample characteristics [7], in the long-standing LBP definition used or in data collection methods [18–20, 42]. In this study, and since no questions were made to participants regarding their LBP symptoms between time-points, it is possible that the persistent group had included people with continuous mild pain with low impact and people with a fluctuating symptom course that varies between severe pain episodes with a large impact on their live and pain-free periods [15].

The results obtained regarding the course of disability and HRQoL seem to support the recurrent pattern of LBP symptoms. Despite statistically significant differences between groups at all follow-up time-points, both groups share a similar recurrent pain pattern over time. For

**Table 3. Odds Ratio (95% Confidence Intervals) for the associations between prognostic indicators and the persistent course of LBP (n = 400).**

| Prognostic indicators | Persistent LBP course | | | | | |
|---|---|---|---|---|---|---|
| | Crude OR | CI 95% | p-value | Adj OR | CI 95% | p-value |
| **Socio-demographic indicators** | | | | | | |
| Age (continuous) | 1.02 | 1.01–1.03 | 0.006 | | | |
| Gender (ref "male") | 2.67 | 1.82–3.93 | <0.001 | 1.90 | 1.26–2.87 | 0.002 |
| Educational level (ref "10 or more years of education") | 1.45 | 0.94–2.22 | 0.093 | | | |
| Marital status (ref "living alone") | 1.10 | 0.79–1.54 | 0.567 | | | |
| **Psychosocial indicators** | | | | | | |
| Anxiety positive symptoms (ref "no") | 1.51 | 1.05–2.17 | 0.025 | | | |
| Depressive positive symptoms (ref "no") | 2.69 | 1.71–4.23 | <0.001 | 1.96 | 1.21–3.18 | 0.007 |
| Employment status (ref "working") | 1.27 | 0.90–1.80 | 0.175 | | | |
| Manual worker (ref "no") | 1.70 | 1.21–2.40 | 0.002 | 1.46 | 1.02–2.10 | 0.040 |
| **Lifestyle indicators** | | | | | | |
| BMI (ref "underweight/normal weight") | 1.44 | 0.99–2.08 | 0.055 | | | |
| Physical exercise (ref "yes") | 1.06 | 0.72–1.58 | 0.756 | | | |
| **Health-related indicators** | | | | | | |
| Pain (question 4- EQ-5D-3L) (ref "moderate pain or discomfort") | 1.53 | 0.93–2.52 | 0.091 | | | |
| Disability (HAQ. 0–3[a]) | 2.19 | 1.71–2.90 | <0.001 | 1.86 | 1.40–2.40 | <0.001 |
| HRQoL (EQ-5D-3L[5] index, 0–1[b]) | 0.74 | 0.03–0.19 | <0.001 | | | |
| **Area Under the Curve (AUC)** | | | | 0.70 | 0.68–0.74 | <0.001 |

Adj, adjusted; BMI, body mass index; EQ-5D-3L, euroqol, 5 dimensions, 3 levels; HAQ, health assessment questionnaire; HRQoL, health-related quality of life; LBP, low back pain; OR, odds ratio; ref, reference.

[a] HAQ, 0–3: A lower HAQ disability score indicates better function.

[b] EQ-5D-3L index, 0–1: "1" corresponds to the best possible health, and "0" to death. Negative values correspond to states of health considered to be worse than death.

example, both groups of patients experienced an improvement in disability between time-points 1 and 2, and a slight worsening between time-points 2 and 3. Nevertheless, it is important to highlight that statistically significant differences found between the baseline and time-point 3 indicate that disability decreases over time in the relapsing group and increases over time in the persistent group. These results challenge both the "one fits all" approach frequently used to manage patients in the real-world clinical setting [43, 44] as well as the recommendations of most clinical guidelines which tend to follow the traditional dichotomy of acute/chronic LBP [45–47]. In the future, it might be of added value to develop and test interventions targeting the different groups of patients with CLBP, namely those aiming to prevent recurrences of pain for patients within a relapsing course, and those focused on promoting long-term self-management for patients within a persistent course.

Especially for the persistent course, its great impact and the worsening pattern on disability and HRQoL strengthen the recommendation for early identification of patients at risk for the development of persistent symptoms, so matched interventions aiming to prevent it may be considered, even for patients with CLBP. In this study, being female, having a manual job, presenting depressive symptoms, and reporting disability increase the chances of a persistent course of LBP. These indicators are consistent with findings reported in other studies [7, 48–51]. Overall, the four indicators identified showed a moderate ability to distinguish between relapsing and persistent courses (AUC 0.70). This result limits the capacity of these indicators to identify at presentation patients who are at risk for persistent, disabling symptoms, and suggests that other potential prognostic factors (e.g., physical, or psychosocial) should be considered to predict persistent LBP.

### Strengths and limitations

Strengths of the current study include its long-term prospective design, the large sample size, data on the long-term impact of persistent versus relapsing LBP courses on disability and HRQoL, and the identification of prognostic indicators for the most impactful course of pain.

Nevertheless, some methodological limitations must be considered. First, only 52.8% of the baseline sample responded to both follow-ups. Although participants included in this study were similar to those lost, except for HRQoL, the possibility of selection bias and residual confounding cannot be ruled out. Another limitation already mentioned was the lack of information regarding health-related outcomes between data collection periods. Furthermore, although positive depressive symptoms were significantly associated with persistent LBP, not all potentially important psychological constructs were included in the present study. For example, fear of pain, catastrophizing [52], and illness perceptions were found to be the strongest predictors of poor prognosis in previous research [53].

Based on this, future studies should seek to enrol patients with an identical starting point of CLBP symptoms' (i.e., inception cohort), follow them more frequently over time, and collect a more comprehensive number of potential predictors to allow a better understanding of distinct courses of pain among patients with CLBP.

## Conclusions

The results of this study provide evidence on the long-term course of back pain and suggested that people with CLBP might have different clinical courses ahead with distinct consequences for disability and HRQoL. Depressive and female patients that have a manual job and report higher levels of disability, have higher chances to develop a persistent course of LBP, the one whose impact on disability and HRQoL is greater.

## Supporting information

**S1 File.**
(XLSX)

## Acknowledgments

The authors' team wishes to acknowledge the contribution of Professor Doctor Carla Nunes for the conceptualization and formal analysis of the current study. In addition, we would like also to acknowledge the EpiDoC Unit and EpiReumaPt team for the invaluable assignment of conceptualization and operationalization of the main research project (EpiDoC), and to CHRC (UIDP/04923/2020) for supporting the current research project.

## Author Contributions

**Conceptualization:** Luís Antunes Gomes, Ana Maria Rodrigues, Jaime C. Branco, Helena Canhão, Eduardo Brazete Cruz.

**Formal analysis:** Luís Antunes Gomes, Ana Maria Rodrigues, Jaime C. Branco, Helena Canhão, Eduardo Brazete Cruz.

**Funding acquisition:** Luís Antunes Gomes, Ana Maria Rodrigues, Jaime C. Branco, Helena Canhão, Eduardo Brazete Cruz.

**Investigation:** Luís Antunes Gomes, Ana Maria Rodrigues, Jaime C. Branco, Helena Canhão, Eduardo Brazete Cruz.

**Methodology:** Luís Antunes Gomes, Ana Maria Rodrigues, Jaime C. Branco, Helena Canhão, Eduardo Brazete Cruz.

**Project administration:** Luís Antunes Gomes, Eduardo Brazete Cruz.

**Supervision:** Ana Maria Rodrigues, Helena Canhão, Eduardo Brazete Cruz.

**Validation:** Luís Antunes Gomes, Ana Maria Rodrigues, Jaime C. Branco, Helena Canhão, Eduardo Brazete Cruz.

**Visualization:** Luís Antunes Gomes, Ana Maria Rodrigues, Eduardo Brazete Cruz.

**Writing – original draft:** Luís Antunes Gomes, Eduardo Brazete Cruz.

**Writing – review & editing:** Luís Antunes Gomes, Ana Maria Rodrigues, Jaime C. Branco, Helena Canhão, Eduardo Brazete Cruz.

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
