## [Decision Letter · Decision Letter 0]

7 Nov 2022

PONE-D-22-05217Female gender, depressive symptoms, manual job, and higher physical disability predict long term low back pain persistencePLOS ONE

Dear Dr. Gomes,

Thank you for submitting your manuscript to PLOS ONE. After careful consideration, we feel that it has merit but does not fully meet PLOS ONE’s publication criteria as it currently stands. Therefore, we invite you to submit a revised version of the manuscript that addresses the points raised during the review process.

We look forward to receiving your revised manuscript.

Kind regards,

Chidozie Emmanuel Mbada, PhD.

Academic Editor

PLOS ONE

Journal Requirements:

“The work presented in this paper is associated to a research project financed by the Portuguese Science & Technology Foundation (PTDC/SAU-SER/7406/2020), awarded by Eduardo Brazete Cruz. The corresponding author, Luís Antunes Gomes, is supported by an individual PhD grant (SFRH/BD/145636/2019) of the same institution.

Portuguese Science & Technology Foundation website: https://www.fct.pt/

EpiReumaPt was supported by unrestricted grants from Direção-Geral da Saúde, Fundação Calouste Gulbenkian, Fundação Champalimaud, Fundação AstraZeneca, Abbvie, Merck, Sharp & Dohme, Pfizer, Roche, Servier, Bial, D3A Medical Systems, Happybrands, Center de Medicina Laboratorial, Germano de Sousa, Clínica Médica da Praia da Vitória, CAL-Clínica, Galp Energia, Açoreana Seguros, and individual rheumatologists. Data were provided by the EpiDoc Unit - CEDOC with permission.

Funders did not play any role in the study design, data collection and analysis, decision to publish, or preparation of the manuscript.”

Additional Editor Comments (if provided):

Dear Authors,

Thank you for submitting your research article to PLoS ONE. The reviewer has recommended major revision, and has made valuable suggestions towards making your manuscript better.

In agreement, I have considered the manuscript both scientifically and technically sound but will need to be re-evaluated after you have attended to the comments by the reviewer. Also, find below additional comments that should be addressed

Abstract section

- Be consistent in writing long-term vs. a long term

- The background statement can be made more succinct. It is too long and less punchy in its present form.

- There should be consistency of language (chronic vs long-term) as used in the abstract and elsewhere. Note that long-term is biopsychosocial model equivalent for the word ‘chronic’. So that the reader is not confusing that with what is meant by long-term in this study, this needs to be clarified.

- It is more appropriate to refer to the population as – ‘patients with active CLBP’ than ‘Active CLBP participants’.

Background

- The objective statement requires recasting to suit the main objective of the study. The part highlighted may be removed or recast, as they are either not directly the focus of this paper or have methodical details that are needed elsewhere in the paper. (The aim of this study was to investigate the presence of different LBP course patterns across 5 years, to examine its cumulative impact on disability and HRQoL and to compare the contribution of socio-demographic, lifestyle, psychosocial and symptom-related indicators to persistent LBP, using data from the EpiDoC).

Materials and Methods

- Include the ethical approval details in the manuscript.

- Delete “or with LBP lasting less than 90 days” in the inclusion/exclusion section as it is an opposite of the inclusion criteria, and really not an exclusion criteria.

Discussion

- The opening sentence (aims statement) needs to be recast (see earlier comment).

Reviewers' comments:

Reviewer's Responses to Questions

**Comments to the Author**

1. Is the manuscript technically sound, and do the data support the conclusions?

Reviewer #1: Partly

2. Has the statistical analysis been performed appropriately and rigorously? 

Reviewer #1: No

3. Have the authors made all data underlying the findings in their manuscript fully available?

Reviewer #1: Yes

4. Is the manuscript presented in an intelligible fashion and written in standard English?

Reviewer #1: No

5. Review Comments to the Author

Reviewer #1: I would like to thank the editor for giving me the chance to review this paper. The paper reports a large prospective study on the impact of persistent LBP on disability and health related quality of life as well as on predictors of long term LBP persistence. The paper reported a study conducted on an important and neglected health condition. Nevertheless, the manuscript has critical editing, organization, and structuring issues. There are lots of language problems as well as conceptual and methodological flaws throughout the paper that needs to be addressed before considering it for publication.

Below are my specific comments, concerns and suggestions in the different sections of the paper.

Abstract

1. The short title is not consistent with the main title. The main title indicates that the focus of the study is to find out the factors which predict long term LBP persistence, whereas, the short title is stated in such a way that the focus of the study is to determine the course and prognosis of low back pain.

2. The main title does not capture the study objective: “to assess the course of chronic LBP (CLBP) over 5 years in a large population-based study, its cumulative impact on disability and health-related quality of life (HRQoL) and the indicators for persistent CLBP course.”

3. It seems that the study is intended to achieve three specific objectives: i) determine the course of CLBP ii) investigate the cumulative impact of CLPB on disability and health related quality of life and iii) find out the indicators for persistent CLBP course. However, the title stated in such a way that the aim of the study is to find out the important factors which predict long term low back pain persistence. Hence, both the main title and the short title need medication in line with the objectives of the study.

4. It is not clear how the participants were selected. The author said that a representative sample of 10, 661 adults were randomly selected from the dwelling population of EpiDoC. But, how the study participants were selected?

5. Page 2, lines 35-36: “A General Linear Model was used to compare mean differences between and within groups.” What are the groups? Repeated measures ANOVA cannot be used to compare differences between groups.

6. The author concluded that “In the long-term, CLBP patients have distinct clinical courses.” What does this mean? No data were presented about clinical course. It seems that data are not presented for all of the three specific objectives the author claims to achieve. For instance, not data were presented on the course of CLBP.

Background

1. The background section is clear, well organized and provides context to the study. It also clearly shows the research gap. However, the aim of the study presented at the end of the background section (Page 6, lines 98-101) is not consistent with the data presented as well as the title of the study. For instance, no one can find data on the “presence of different LBP course patterns.”

2. The background section does not present synthesis of the existing evidence and research gaps related to the impact of CLBP on disability and HRQL and factors that predict CLBP persistence. What previous studies say with regard to the above research questions and what are the research gaps?

Methods

1. The author indicated that “The detailed procedures for the study design and sample selection and recruitment are published elsewhere” (Page 6, lines 110-111). This is Ok but there is need to briefly describe the study design and sample selection procedure as this paper has to be standalone. Need also to have “study design” and “sample selection” sub-sections.

2. The author needs to describe in detail the outcome measures of the Health Assessment Questionnaire (HAQ) and the EQ-5D-3L (number of items, content of items, psychometric properties and their appropriateness to the Portuguese situation). There is also need to provide more information about the HADS scale which was used to measure depression and anxiety.

3. It is not clear why disability and HRQL are considered as symptom-related indicators? Page 8, line 153.

4. The statement “Secondly, and as different CLBP courses were expected, descriptive statistics were planned for different CLBP groups” (Page 9, lines 167-168) is not clear at all.

5. Page 9, lines 170-171: “Differences between groups on disability and HRQoL, over time, were examined using repeated measures analysis of variance.” What are these groups? Repeated measures ANOVA assumes that the outcome variable (i. e. disability and HRQL in this case) needs to be normality distributed. Not clear whether or not this was checked.

6. What statistical method was used to analyze data to address one of the objectives of the study (i. e. to investigate the presence of different LBP course patterns across 5 years?

Results

1. Page 10, lines 195-196: “At baseline, no statistically significant differences in socio-demographic and clinical characteristics were found between responders and non-responders” Who are the responders and who are the non-responders at baseline? Do you mean those who had LBP and those who had not?

2. The age for the total sample is not reported (in Table 1). Why?

3. While the frequency for each category of educational level is reported, the percentage in brackets is not reported. Why?

4. Page 12, lines 233-235: “At any time-point there were statistically significant differences on disability (p<0.005) and HRQoL (EQ-5D-3L index, p<0.005), between groups.” It is possible to know this with a simple look at the descriptive data. However, it is not clear what statistical method was used to test this. Repeated measures ANOVA can only test within group differences, but not between group differences.

5. The statement “HRQoL statistically significantly decreased (p<0.001) and disability significantly increased (p<0.001) in the persistent group” (Page 13, lines 238-239) does not seem to be correct as the data on Table 2 doesn’t show the same.

Discussion

1. Overall, the discussion section is not well written and the major findings of the study are not properly interpreted. What the findings mean to the context where the study was conducted, and to specific nature of the sample? Findings are also not well related to previous literature, theory and treatment guidelines.

2. The implications of the findings to clinical practice and policy making are not described. Future directions to research in the area also are not indicated.

6. PLOS authors have the option to publish the peer review history of their article (what does this mean?). If published, this will include your full peer review and any attached files.

Reviewer #1: **Yes: **Kassahun Habtamu

---

## [Author Response · Author response to Decision Letter 0]

21 Jan 2023

Academic Editor Doctor Chidozie Emmanuel Mbada,

RESPONSE: The manuscript was revised and updated accordingly to the PLOS ONEs’ style requirements.

“The work presented in this paper is associated to a research project financed by the Portuguese Science & Technology Foundation (PTDC/SAU-SER/7406/2020), awarded by Eduardo Brazete Cruz. The corresponding author, Luís Antunes Gomes, is supported by an individual PhD grant (SFRH/BD/145636/2019) of the same institution.

Portuguese Science & Technology Foundation website: https://www.fct.pt/

EpiReumaPt was supported by unrestricted grants from Direção-Geral da Saúde, Fundação Calouste Gulbenkian, Fundação Champalimaud, Fundação AstraZeneca, Abbvie, Merck, Sharp & Dohme, Pfizer, Roche, Servier, Bial, D3A Medical Systems, Happybrands, Center de Medicina Laboratorial, Germano de Sousa, Clínica Médica da Praia da Vitória, CAL-Clínica, Galp Energia, Açoreana Seguros, and individual rheumatologists. Data were provided by the EpiDoc Unit - CEDOC with permission.

Funders did not play any role in the study design, data collection and analysis, decision to publish, or preparation of the manuscript.”

RESPONSE: As suggested by the Editor, an amendment to the Funding Statement was provided in the cover letter that accompanies this review. In addition to this, the name of the unit providing data was also updated to match the recent re-organization of our research unit.

“Data were provided by the EpiDoC Unit, NOVA Medical School, Universidade NOVA de Lisboa with permission”

“There was no additional external or internal funding received for this study.”

RESPONSE: The ethical approval details were included in the materials and methods section (first paragraph), as suggested by the Editor.

“The EpiDoC followed the principles established by the Declaration of Helsinki and revised in 2013 in Fortaleza. The study was approved by the Ethics Committee of NOVA Medical School and the Portuguese Data Protection Authority. All participants gave written informed consent [22,23].”

RESPONSE: A new section was added after the conclusions section based on the Editors’ comment.

Supporting information

S1 Dataset (XLSX)

Additional Editor Comments:

Abstract section

- Be consistent in writing long-term vs. a long term

RESPONSE: Based on the Editors’ suggestion, the manuscript text was revised and changed accordingly.

- The background statement can be made more succinct. It is too long and less punchy in its present form.

RESPONSE: Based on the Editors’ suggestion, the background section of the abstract was made more succinct.

“Background: Low back pain (LBP) is a long-term health condition with distinct clinical courses. Its characterization together with the identification of prognostic factors for a persistent LBP course may trigger the development of personalized interventions. This study aimed to investigate the courses of chronic LBP (CLBP), its cumulative impact, and the indicators for the persistence of pain.”

- There should be consistency of language (chronic vs long-term) as used in the abstract and elsewhere. Note that long-term is biopsychosocial model equivalent for the word ‘chronic’. So that the reader is not confusing that with what is meant by long-term in this study, this needs to be clarified.

RESPONSE: Based on the Editors’ suggestion, the text was revised and updated accordingly. The term chronic was used only to describe the diagnosis of the participants.

- It is more appropriate to refer to the population as – ‘patients with active CLBP’ than ‘Active CLBP participants’.

RESPONSE: Based on the Editors’ suggestion, the text was revised and changed accordingly.

Background

- The objective statement requires recasting to suit the main objective of the study. The part highlighted may be removed or recast, as they are either not directly the focus of this paper or have methodical details that are needed elsewhere in the paper. (The aim of this study was to investigate the presence of different LBP course patterns across 5 years, to examine its cumulative impact on disability and HRQoL and to compare the contribution of socio-demographic, lifestyle, psychosocial and symptom-related indicators to persistent LBP, using data from the EpiDoC).

RESPONSE: We agree and thank for the Reviewer's suggestion. The title, objectives (last paragraph of the background section), and material and methods sections were revised to relate them which each other in a more direct way.

Full title: “Clinical courses, impact and prognostic indicators for a persistent course of low back pain: results from a population-based cohort study”

“The aims of this study were 1) to investigate the presence of different courses of pain across five years in a sample of patients with active CLBP; 2) to examine their cumulative impact on disability and HRQoL and; 3) to identify prognostic indicators for the persistent LBP course.”

Materials and Methods

- Include the ethical approval details in the manuscript.

RESPONSE: The ethical approval details were included in the materials and methods section (first paragraph), as suggested by the Editor.

“The EpiDoC followed the principles established by the Declaration of Helsinki and revised in 2013 in Fortaleza. The study was approved by the Ethics Committee of NOVA Medical School and the Portuguese Data Protection Authority. All participants gave written informed consent [22,23].”

- Delete “or with LBP lasting less than 90 days” in the inclusion/exclusion section as it is an opposite of the inclusion criteria, and really not an exclusion criteria.

RESPONSE: The information “or with LBP lasting less than 90 days” was deleted as suggested by the Editor.

Discussion

- The opening sentence (aims statement) needs to be recast (see earlier comment).

RESPONSE: The opening sentence of the discussion section was rectified according to the Editors’ suggestion and to better match the change on both the full title, aims (introduction section) and materials and methods section.

“This study aimed to identify different courses of pain within a sample of patients with active CLBP during a follow-up period of five years; to examine their cumulative impact on disability and HRQoL and; to identify potential predictors for the persistence of symptoms.”

 

Reviewer Doctor Kassahun Habtamu,

Abstract 

1. The short title is not consistent with the main title. The main title indicates that the focus of the study is to find out the factors which predict long term LBP persistence, whereas, the short title is stated in such a way that the focus of the study is to determine the course and prognosis of low back pain.

RESPONSE: We agree and thank for the Reviewers’ comment. Both the full and short titles were rectified to better connect them, accordingly to the Editors’ and Reviewers’ suggestions. 

Full title: “Clinical courses, impact and prognostic indicators for a persistent course of low back pain: results from a population-based cohort study”

Short title: “Course, impact, and prognosis of low back pain persistence”

2. The main title does not capture the study objective: “to assess the course of chronic LBP (CLBP) over 5 years in a large population-based study, its cumulative impact on disability and health-related quality of life (HRQoL) and the indicators for persistent CLBP course.”

RESPONSE: We agree and thank for the Reviewers’ comment. To better connect both the full title and aims of the study (abstract and last paragraph of introduction), both sections were rectified.

Full title: “Clinical courses, impact and prognostic indicators for a persistent course of low back pain: results from a population-based cohort study”

“This study aimed to investigate the courses of chronic LBP (CLBP), its cumulative impact, and the indicators for the persistence of pain.”

“The aims of this study were 1) to investigate the presence of different courses of pain across five years in a sample of patients with active CLBP; 2) to examine their cumulative impact on disability and HRQoL and; 3) to identify prognostic indicators for the persistent LBP course.”

3. It seems that the study is intended to achieve three specific objectives: i) determine the course of CLBP ii) investigate the cumulative impact of CLPB on disability and health related quality of life and iii) find out the indicators for persistent CLBP course. However, the title stated in such a way that the aim of the study is to find out the important factors which predict long term low back pain persistence. Hence, both the main title and the short title need medication in line with the objectives of the study.

RESPONSE: We agree that the relationship between the full and short title and study aims was not as clear as it seems. Thus, we appreciated the Reviewers’ comments. To better and more directly connect both the full and short title with the aims of the study (abstract and last paragraph of introduction), all these sections were updated.

Full title: “Clinical courses, impact and prognostic indicators for a persistent course of low back pain: results from a population-based cohort study”

Short title: “Course, impact, and prognosis of low back pain persistence”

“This study aimed to investigate the courses of chronic LBP (CLBP), its cumulative impact, and the indicators for the persistence of pain.”

“The aims of this study were 1) to investigate the presence of different courses of pain across five years in a sample of patients with active CLBP; 2) to examine their cumulative impact on disability and HRQoL and; 3) to identify prognostic indicators for the persistent LBP course.”

4. It is not clear how the participants were selected. The author said that a representative sample of 10, 661 adults were randomly selected from the dwelling population of EpiDoC. But, how the study participants were selected?

RESPONSE: The present study was conducted under the scope of the EpiDoC, a population-based study with a prolonged follow-up which started with the EpiReumaPt (EpiDoC 1). EpiReumaPt was a national, cross-sectional, population-based study aiming to estimate the prevalence of rheumatic and musculoskeletal diseases in the adult Portuguese population (>18 years old). To collect a representative sample of the Portuguese population, the selection of patients was guided by a three-stage approach which started with face-to-face interviews to randomly select individuals’ households. Households were selected based on a random route methodology following a multistage random sampling, taking into account the stratification of the Portuguese statistical administrative territorial units (Nomenclature of Territorial Units for Statistics) in the 2001 CENSUS and the size of the population. This information was summarized and added to the methods and materials section of the abstract. The full description of the selection process can be now found in the materials and methods section.

“Participants with active CLBP from the EpiDoC, a population-based cohort study of a randomly recruited sample of 10.661 adults with prolonged follow-up, were considered. Pain, disability, and health-related quality of life (HRQoL) were assessed at three time-points over five years.”

5. Page 2, lines 35-36: “A General Linear Model was used to compare mean differences between and within groups.” What are the groups? Repeated measures ANOVA cannot be used to compare differences between groups.

RESPONSE: Patients with CLBP were divided regarding their clinical course of pain into two groups: “Persistent LBP course” and “Relapsing LBP course”. Each group was repeatedly evaluated regarding their disability and HRQoL over three time-points. To test both the within and between groups differences, a mixed ANOVA (General Linear Model procedure) was used. In this test, our "within-group" factor was the time (disability measured with HAQ at time-points 1, 2 and 3; and HRQoL measured with EQ-5D-3L index at time-points 1, 2 and 3 – dependent variables); while our “between-group” factor was the courses of LBP (independent variable). To clarify this procedure, the text was revised and updated accordingly in the material and methods section of the abstract.

“A mixed ANOVA was used to compare mean differences within and between patients of distinct courses.”

6. The author concluded that “In the long-term, CLBP patients have distinct clinical courses.” What does this mean? No data were presented about clinical course. It seems that data are not presented for all of the three specific objectives the author claims to achieve. For instance, not data were presented on the course of CLBP.

RESPONSE: Based on previous literature, two main courses of LBP were expected to be found: a persistent course where patients report pain in all follow-up time-points, and a relapsing course where patients report pain-free moments. The methods used to identify these courses, and the results of this analysis were summarized in the abstract section (material and methods, results, and conclusion section), accordingly to the Editor and Reviewers’ suggestion. 

“According to their pain symptoms over time, participants were classified as having a persistent (pain at the baseline and at all the subsequent time-points) or a relapsing pain course (pain at the baseline and no pain at least in one of the subsequent time-points).”

“Among the 1.201 adults with active CLBP at baseline, 634 (52.8%) completed the three time-points of data collection: 400 (63.1%) had a persistent and 234 (36.9%) a relapsing course.”

“In the long-term, patients with CLBP may follow a persistent or relapsing course of pain.”

Background 

1. The background section is clear, well organized and provides context to the study. It also clearly shows the research gap. However, the aim of the study presented at the end of the background section (Page 6, lines 98-101) is not consistent with the data presented as well as the title of the study. For instance, no one can find data on the “presence of different LBP course patterns.”

RESPONSE: The aims of the study were revised and written more clearly and directly. Additionally, some revisions at the nomenclature level were performed in the background section to better introduce the previous literature on courses of LBP (fourth paragraph).

“The population-based studies with long-term follow-up (5 to 10 years) published to date have been contesting the traditional classification of LBP, suggesting that more than an episodic condition of unrelated episodes, LBP is a long-term condition with a variable course of pain and multiple interrelated episodes [14–21]. Despite differences in the case definition or in the assessment time-points, only 6 to 14% of adults enrolled in these studies have a persistent course of LBP, with pain at all specific time-points of the follow-up [18,19].”

“The aims of this study were 1) to investigate the presence of different courses of pain across five years in a sample of patients with active CLBP; 2) to examine their cumulative impact on disability and HRQoL and; 3) to identify prognostic indicators for the persistent LBP course.”

2. The background section does not present synthesis of the existing evidence and research gaps related to the impact of CLBP on disability and HRQL and factors that predict CLBP persistence. What previous studies say with regard to the above research questions and what are the research gaps?

RESPONSE: The research gap this study intends to fill is related with the identification of prognostic indicators for the most burdensome course of LBP, the persistent course. Despite the existence of previous literature on this topic, most studies have single or short-term follow-up timepoints or/and are based on a heterogeneous population of patients with LBP which are very important limitations considering the life-lasting condition pattern of LBP. To address these limitations, the present study focused its research question on 1) long-term and repeated follow-up time-points (5 years) and 2) patients with chronic LBP which are a minority subgroup of patients for which the impact of LBP is great (paragraph two of background section). Based on the Reviewers’ suggestion, we clarified this novelty in the background section (fourth and fifth paragraphs).

“… specific time-points, making it difficult...”

“The population-based studies with long-term follow-up (5 to 10 years) published to date have been contesting the traditional classification of LBP, suggesting that more than an episodic condition of unrelated episodes, LBP is a long-term condition with a variable course of pain and multiple interrelated episodes [14–21].”

“…Although a set of indicators has been associated with the development of persistent, disabling pain such as disability, low levels of HRQoL, maladaptive pain coping behaviours, or depressive symptoms [25–30], the majority have emerged from studies that have evaluated pain at a single point of time, with follow-up limited to one year and based on heterogenous samples of patients regarding the previous history of LBP [31,32].”

Methods 

1. The author indicated that “The detailed procedures for the study design and sample selection and recruitment are published elsewhere” (Page 6, lines 110-111). This is Ok but there is need to briefly describe the study design and sample selection procedure as this paper has to be standalone. Need also to have “study design” and “sample selection” sub-sections.

RESPONSE: We agree and thank for the Reviewer's suggestion. Both the study design and sample selection were described in the materials and methods section of the manuscript, under the new subsection “Data source and study design” (first paragraph). Additionally, for better understanding of this procedure, figure 1 describing the flow diagram of recruitment (results section) was also updated.

“This study used longitudinal data of participants with active CLBP collected under the scope of the EpiDoC, a prospective cohort study that began with the EpiDoC 1 (EpiReumaPt) in 2011-13. EpiReumaPt was a national, cross-sectional, population-based study with a representative sample of the Portuguese population, comprising 10.661 non-institutionalized adults (≥18 years), living in Portugal and able to speak and read the Portuguese language [22,23]. The recruitment of participants followed a three-stage approach. Aiming to characterize the Portuguese adult population and screen for the presence of rheumatic and musculoskeletal diseases (RMDs), the selection of participants for the first stage of the study (RMDs screening) followed a multistage random sampling, taking into account the stratification of the Portuguese statistic administrative territorial units (Nomenclature of Territorial Units for Statistics) in the 2001 CENSUS and the size of the population. Using a random route methodology, face-to-face interviews to randomly select individuals’ households were conducted by a team of interviewers (i.e., non-physicians trained for this purpose). In the second stage (RMD diagnosis), both individuals who screened positive for at least one RMD and a random sample of 20% of individuals free of RMDs were evaluated by a rheumatologist so a diagnosis could be established. In the third and last stage (RMDs diagnostic validation), a team of experienced rheumatologists validated the previous RMD diagnosis. All participants recruited for the first stage of EpiDoC 1 were invited to participate in the following waves of the study; 10.153 (95.2%) agreed and signed the consent forms. From there, and over a 5-year period, two subsequent time-points of data collection (every 18 months) have been completed, through telephone interviews, using data based on the same participants [24]. To prevent withdrawals, when a participant was not available, a maximum of 6 additional attempts were made at different times [24]. The detailed procedures for the study design and sample selection and recruitment are published elsewhere [23,24].”

2. The author needs to describe in detail the outcome measures of the Health Assessment Questionnaire (HAQ) and the EQ-5D-3L (number of items, content of items, psychometric properties and their appropriateness to the Portuguese situation). There is also need to provide more information about the HADS scale which was used to measure depression and anxiety.

RESPONSE: We agree and thank for the Reviewer's suggestion. Information about HAQ, EQ-5D-3L, and HADS was detailed in the “outcome measures” and “prognostic indicators” subsections of the materials and methods section. 

“Disability was measured using the Portuguese version of the Health Assessment Questionnaire (HAQ), a 20-question instrument that evaluates eight categories of functioning (dressing, rising, eating, walking, hygiene, reach, grip, and usual activities) over the previous week. Each category was scored from 0 (no disability) to 3 points (completely disabled). The final score also varied between 0 and 3 points, with higher scores corresponding to higher disability [33]. HRQoL was measured with the Portuguese version of EQ-5D-3L. This instrument comprises five questions regarding different dimensions of HRQoL (mobility, self-care, usual activities, pain/ discomfort, and anxiety/ depression) which are scored on a three-level severity scale (0= “no problem”; 1= “moderate problem”, or 2= “extreme problem”). The overall index of health status ranges between 0 (“death”) and 1 (“the best possible health”). The weight applied to the severity states of the EQ-5D-3L was based on the Portuguese valuation study of the EQ-5D-3L [34]. Both HAQ and EQ-5D-3L instruments have been cross-culturally validated for the Portuguese language and showed good to excellent psychometric properties [34,35].”

“…The HADS has been cross-culturally validated in the Portuguese language and showed good to excellent psychometric properties [37,38].”

3. It is not clear why disability and HRQL are considered as symptom-related indicators? Page 8, line 153.

RESPONSE: Indeed, “symptom-related indicators” are not the most correct classification for disability and HRQoL outcomes. Based on the Reviewers’ comments, this term was replaced by “health-related indicators” in the prognostic indicators subsection of the materials and methods section, and in Table 3 of the results section.

“…and health-related indicators (disability and HRQoL).”

4. The statement “Secondly, and as different CLBP courses were expected, descriptive statistics were planned for different CLBP groups” (Page 9, lines 167-168) is not clear at all.

RESPONSE: Based on the lack of clarity throughout the manuscript text associated with the investigation of different courses of LBP, this procedure was further developed in the statistical analysis of methods section (third paragraph).

“Secondly, different courses of LBP were represented by a three-word variable based on the participants’ answer to question 4 of the EQ-5D-3L: from left to right, the first word stands for pain status at baseline, while the second and the third for the pain status at time-point 2 and 3, respectively. Since every participant had “moderate or extreme pain/discomfort” (pain dimension of the EQ-5D-3L) at baseline, courses of pain were defined based on the pain status of subsequent time-points. Two different courses were expected: a “persistent LBP course” for participants who reported “moderate or extreme pain/discomfort” in both time-points 2 and 3 (pain/pain/pain); and a “relapsing course” for those reporting “no pain/discomfort” at least in one time-point (pain/pain/no pain; pain/no pain/pain or; pain/no pain/no pain).”

5. Page 9, lines 170-171: “Differences between groups on disability and HRQoL, over time, were examined using repeated measures analysis of variance.” What are these groups? Repeated measures ANOVA assumes that the outcome variable (i. e. disability and HRQL in this case) needs to be normality distributed. Not clear whether or not this was checked.

RESPONSE: As stated in a previous comment response, aiming to identify and characterize distinct courses of pain in patients with CLBP, our sample was divided into two different groups: those with a “persistent LBP course” and those with a “relapsing LBP course”. Each group was repeatedly evaluated regarding their disability and HRQoL over three time-points. Since we were interested in investigating not only the within but also the between-group differences regarding disability and HRQoL, a mixed ANOVA (General Linear Model procedure) was used. In this test, our "within-group" factor was the time (disability measured with HAQ at time-points 1, 2 and 3; and HRQoL measured with EQ-5D-3L index at time-points 1, 2 and 3 – dependent variables); while our “between-group” factor was the courses of LBP (independent variable). To clarify this procedure, the text of the statistical analysis subsection of the material and methods section was clarified.

“Within and between groups differences in disability and HRQoL, over the three time-points, were examined using a mixed ANOVA (General Linear Model procedure). Aiming to identify which pairs of means differed from each other, for the interaction between groups and time-points, a post-hoc test comprising multiple comparison testing with Bonferroni correction was performed.”

Regarding the normality of the outcomes, we understand the Reviewer's suggestion although given the large sample used and based on the Central Limit Theorem which suggests that “for a large sample of n observations from a population with a finite mean and variance, the sampling distribution of the sum or mean of samples of size n is approximately normal”, we assumed the normality of our data.

Kwak SG, Kim JH. Introduction Basic Concepts of Central Limit Theorem Central limit theorem: the cornerstone of modern statistics KJA. Korean J Anesthesiol 2017;70:144–56. doi:10.4097/kjae.2017.70.2.144

6. What statistical method was used to analyze data to address one of the objectives of the study (i. e. to investigate the presence of different LBP course patterns across 5 years? 

RESPONSE: The statistical methods used to investigate the different courses of LBP was further developed in the statistical analysis of methods section (third paragraph).

“Secondly, different courses of LBP were represented by a three-word variable based on the participants’ answer to question 4 of the EQ-5D-3L: from left to right, the first word stands for pain status at baseline, while the second and the third for the pain status at time-point 2 and 3, respectively. Since every participant had “moderate or extreme pain/discomfort” (pain dimension of the EQ-5D-3L) at baseline, courses of pain were defined based on the pain status of subsequent time-points. Two different courses were expected: a “persistent LBP course” for participants who reported “moderate or extreme pain/discomfort” in both time-points 2 and 3 (pain/pain/pain); and a “relapsing course” for those reporting “no pain/discomfort” at least in one time-point (pain/pain/no pain; pain/no pain/pain or; pain/no pain/no pain).”

Results 

1. Page 10, lines 195-196: “At baseline, no statistically significant differences in socio-demographic and clinical characteristics were found between responders and non-responders” Who are the responders and who are the non-responders at baseline? Do you mean those who had LBP and those who had not? 

RESPONSE: For nonresponse bias assessment, responders were considered all participants who fulfilled the eligibility criteria (i.e., having CLBP, baseline pain when answering question 4 of the EQ-5D-3L, and participated in all the time-points of data collection), whereas non-responders were those who, despite the presence of both CLBP and baseline pain when answering to question 4 of the EQ-5D-3L, had not participated in all time-points of data collection. Reasons for not participating were described in the first paragraph of the results section.

Based on the Reviewers’ question, this information was added to the statistical analysis subsection of the materials and methods section. 

“… For this analysis, responders were those who fulfilled the eligibility criteria, while non-responders were those who, despite the presence of both CLBP and baseline pain measured by question 4 of the EQ-5D-3L, had not participated in all time-points of data collection.”

2. The age for the total sample is not reported (in Table 1). Why? 

RESPONSE: The age of total sample was imbibed in the text (second paragraph of the results section). Additionally, following Reviewers’ suggestion, it was also added to Table 1. In line with this update, we also added the mean ± SD value of HAQ and EQ-5D-3L.

“Age (years), mean ± SD for total sample = 61.6 ± 13.6”

“Disability (HAQ, 0-3),a mean ± SD (n=628) = 0.99 ± 0.72”

“HRQoL (EQ-5D-3L index, 0-1),b mean ± SD = 0.46 ± 0.19”

3. While the frequency for each category of educational level is reported, the percentage in brackets is not reported. Why? 

RESPONSE: By mistake, the frequency of each category of the educational level was not described in Table 1. During this update, some “copy paste” inaccuracies were detected in other baseline sociodemographic variables. Because of this, table 1 was revised and updated.

4. Page 12, lines 233-235: “At any time-point there were statistically significant differences on disability (p<0.005) and HRQoL (EQ-5D-3L index, p<0.005), between groups.” It is possible to know this with a simple look at the descriptive data. However, it is not clear what statistical method was used to test this. Repeated measures ANOVA can only test within group differences, but not between group differences. 

RESPONSE: We would like to take this opportunity to clarify that we did not use a repeated measures ANOVA, but instead a mixed ANOVA was performed. Despite both tests aiming to compare mean differences between two factors, a repeated measures ANOVA has two “within group” factors and thus does not allow testing between group interaction, as stated by the Reviewer. In contrast, a mixed ANOVA involves the same two factors, although one is “within group" and the other is "between group ". This test allowed us to test both the within and between-group differences. Despite the suggestion of between-group differences in disability and HRQoL with a simple look at table 2, we insisted on testing these differences through statistical tests designed specifically for situations like these. To clarify this procedure, the text of the statistical analysis subsection of the material and methods section was clarified.

“Within and between groups differences in disability and HRQoL, over the three time-points, were examined using a mixed ANOVA (General Linear Model procedure). Aiming to identify which pairs of means differed from each other, for the interaction between groups and time-points, a post-hoc test comprising multiple comparison testing with Bonferroni correction was performed.”

5. The statement “HRQoL statistically significantly decreased (p<0.001) and disability significantly increased (p<0.001) in the persistent group” (Page 13, lines 238-239) does not seem to be correct as the data on Table 2 doesn’t show the same. 

RESPONSE: This interpretation was based on the self-reported instruments used to evaluate HRQoL and disability. HRQoL was evaluated with the EQ-5D-3L whose overall index ranges between 0 and 1; higher scores suggest better HRQoL. Regarding disability, it was evaluated using the HAQ whose score ranges between 0 and 3; higher scores correspond to higher disability. All the comparisons between timepoints were statistically significant (p< .001).

Table 2 shows that for the relapsing LBP course, the EQ-5D-3L index was higher for time-point 3 (0.70 ±0.32) compared with time-point 1 (0.52 ±0.19), suggesting an improvement in HRQoL. For disability, the HAQ score was lower for time-point 3 (0.55 ±0.60) compared with time-point 1 (0.76 ±0.69), suggesting an improvement in disability.

Regarding the persistent course, the EQ-5D-3L index was lower for time-point 3 (0.34 ±0.24) compared with time-point 1 (0.43 ±0.19), suggesting a decrease in HRQoL. For disability, the HAQ score was higher for time-point 3 (1.27 ±0.70) compared with time-point 1 (1.13 ±0.70), suggesting an improvement in disability. All the comparisons between timepoints were statistically significant (p<0.001).

Since the words “decreased” and “increased” can be confusing, they were replaced by the words “higher” and “lower”, to better match the previous formulation in the respective phrase (“Course of disability and HRQoL between groups, overtime” subsection of the results section). The word “overtime”, in the beginning of this phrase was also deleted.

“Moreover, an opposite pattern of impact was found between groups; compared to baseline, HRQoL was significantly higher (EQ-5D-3L, p<0.001), and disability was significantly lower (HAQ, p<0.001) at time-point 3 in the relapsing group, whereas HRQoL was significantly lower (EQ-5D-3L, p<0.001), and disability was significantly higher (HAQ, p<0.001) at time-point 3 in the persistent group.”

Discussion 

1. Overall, the discussion section is not well written and the major findings of the study are not properly interpreted. What the findings mean to the context where the study was conducted, and to specific nature of the sample? Findings are also not well related to previous literature, theory and treatment guidelines. 

RESPONSE: Based on the Reviewers’ feedback, the discussion section was generally revised, making it more focused on the clinical context and the nature of CLBP. 

2. The implications of the findings to clinical practice and policy making are not described. Future directions to research in the area also are not indicated. 

RESPONSE: Implications of the findings to clinical practice and policy, as well as recommendations for future studies, were added to the discussion section (last two paragraphs and “strengths and limitation” section)

“These results challenge both the “one fits all” approach frequently used to manage patients in the real-world clinical setting [43,44] as well as the recommendations of most clinical guidelines which tend to follow the traditional dichotomy of acute/chronic LBP [45–47]. In the future, it might be of added value to develop and test interventions targeting the different groups of patients with CLBP, namely those aiming to prevent recurrences of pain for patients within a relapsing course, and those focused on promoting long-term self-management for patients within a persistent course.”

“Especially for the persistent course, its great impact and the worsening pattern on disability and HRQoL strengthen the recommendation for early identification of patients at risk for the development of persistent symptoms, so matched interventions aiming to prevent it may be considered, even for patients with CLBP.”

“Based on this, future studies should seek to enrol patients with an identical starting point of CLBP symptoms’ (i.e., inception cohort), follow them more frequently over time, and collect a more comprehensive number of potential predictors to allow a better understanding of distinct courses of pain among patients with CLBP.”

---

## [Decision Letter · Decision Letter 1]

6 Mar 2023

Clinical courses, impact and prognostic indicators for a persistent course of low back pain: results from a population-based cohort study

PONE-D-22-05217R1

Dear Dr. Gomes,

We’re pleased to inform you that your manuscript has been judged scientifically suitable for publication and will be formally accepted for publication once it meets all outstanding technical requirements.

Kind regards,

Chidozie Emmanuel Mbada, PhD.

Academic Editor

PLOS ONE

Additional Editor Comments (optional):

Dear Authors,

We now have comments from two reviewers on your resubmission. Both reviewers seem assured with the improvement made so far to the manuscript. In accordance with their positions, I advise that the pending minor revision suggested by reviewer #2 be addressed.

Reviewers' comments:

Reviewer's Responses to Questions

**Comments to the Author**

1. If the authors have adequately addressed your comments raised in a previous round of review and you feel that this manuscript is now acceptable for publication, you may indicate that here to bypass the “Comments to the Author” section, enter your conflict of interest statement in the “Confidential to Editor” section, and submit your "Accept" recommendation.

Reviewer #1: All comments have been addressed

Reviewer #2: (No Response)

2. Is the manuscript technically sound, and do the data support the conclusions?

Reviewer #1: Yes

Reviewer #2: Yes

3. Has the statistical analysis been performed appropriately and rigorously? 

Reviewer #1: (No Response)

Reviewer #2: No

4. Have the authors made all data underlying the findings in their manuscript fully available?

Reviewer #1: (No Response)

Reviewer #2: Yes

5. Is the manuscript presented in an intelligible fashion and written in standard English?

Reviewer #1: Yes

Reviewer #2: Yes

6. Review Comments to the Author

Reviewer #1: Thank you for giving me the chance for reviewing this paper again.

The authors addressed all the comments I provided in the earlier version of the manuscript and has now been significantly improved. I have no any further comments. The paper can now be accepted for publication.

Reviewer #2: (No Response)

7. PLOS authors have the option to publish the peer review history of their article (what does this mean?). If published, this will include your full peer review and any attached files.

Reviewer #1: **Yes: **Kassahun Habtamu

Reviewer #2: No

---

## [Editor Report · Acceptance letter]

9 Mar 2023

PONE-D-22-05217R1 

Clinical courses, impact and prognostic indicators for a persistent course of low back pain: results from a population-based cohort study 

Dear Dr. Gomes:

I'm pleased to inform you that your manuscript has been deemed suitable for publication in PLOS ONE. Congratulations! Your manuscript is now with our production department. 

Kind regards, 

on behalf of

Dr. Chidozie Emmanuel Mbada 

Academic Editor

PLOS ONE